# Molecular Diversity of Hard Tick Species from Selected Areas of a Wildlife-Livestock Interface Ecosystem at Mikumi National Park, Morogoro Region, Tanzania

**DOI:** 10.3390/vetsci8030036

**Published:** 2021-02-24

**Authors:** Donath Damian, Modester Damas, Jonas Johansson Wensman, Mikael Berg

**Affiliations:** 1Department of Molecular Biology and Biotechnology, University of Dar es Salaam, 35091 Dar es Salaam, Tanzania; damas.modester@gmail.com; 2Department of Biomedical Sciences and Veterinary Public Health, Swedish University of Agricultural Sciences, Box 7036, SE-750 07 Uppsala, Sweden; mikael.berg@slu.se; 3Department of Clinical Sciences, Swedish University of Agricultural Sciences, Box 7054, SE-750 07 Uppsala, Sweden; jonas.wensman@slu.se

**Keywords:** hard ticks, 16S rRNA gene, *Hyalomma*, *Rhipicephalus*, wildlife-livestock interface, Mikumi National Park

## Abstract

Ticks are one of the most important arthropod vectors and reservoirs as they harbor a wide variety of viruses, bacteria, fungi, protozoa, and nematodes, which can cause diseases in human and livestock. Due to their impact on human, livestock, and wild animal health, increased knowledge of ticks is needed. So far, the published data on the molecular diversity between hard ticks species collected in Tanzania is scarce. The objective of this study was to determine the genetic diversity between hard tick species collected in the wildlife-livestock interface ecosystem at Mikumi National Park, Tanzania using the mitochondrion 16S rRNA gene sequences. Adult ticks were collected from cattle (632 ticks), goats (187 ticks), and environment (28 ticks) in the wards which lie at the border of Mikumi National Park. Morphological identification of ticks was performed to genus level. To identify ticks to species level, molecular analysis based on mitochondrion 16S rRNA gene was performed. Ticks representing the two genera (*Hyalomma* and *Rhipicephalus*) were identified using morphological characters. Six species were confirmed based on mitochondrion 16S rRNA gene, including *Rhipicephalus microplus*, *Rhipicephalus evertsi*, *Hyalomma rufipes*, *Hyalomma truncatum*, *Hyalomma marginatum*, and *Hhyalomma turanicum*. The presence of different clusters of tick species reflects the possible biological diversity of the hard ticks present in the study region. Further studies are however required to quantify species of hard ticks present in the study region and the country in general over a larger scale.

## 1. Introduction

Ticks are responsible for great economic, social, and conservation losses because of their negative impacts on human, livestock, and wild animal health [1]. Ticks can cause reduction in animal body weight, limit animal production and induce anemia especially when animal is heavily infested [2]. Tick bites may cause irritation resulting in reduced quality of the hides [2]. Certain tick species, *Rhipicephalus evertsi*, *Ixodes rubicundus*, and *Hyalomma truncatum* inject toxin to animals, which causes paralysis [1]. Ticks are also reservoirs and vectors of viruses, bacteria, fungi, protozoa, and nematodes, which can cause diseases in human, livestock, and wild animals [3].

Most of the tick species belong to the two main families, *Ixodidae* (hard ticks) and *Argasidae* (soft ticks) [4,5]. Hard ticks of the genera *Rhipicephalus*, *Hyalomma*, and *Amblyomma* are the most important and widely distributed species found in many parts of Africa, including Tanzania [6,7,8].

More than 95% of the cattle and goat population in Tanzania are reared under traditional pastoral systems [9]. Nomadic and pastoralist lifestyles, especially those at the wildlife-livestock interface, can allow direct and indirect contact with wild animals that can facilitate exposure and sharing of tick spp., [1].

In areas which lie at the border of Mikumi National Park, people practice nomadic pastoralism, keeping large number of indigenous cattle and goats. During dry seasons there is the movement of people and livestock to areas very close and sometimes entering beyond the National Park boundary, where water and pasture are abundant long after the rains have gone. Likewise, there is migration of wild animals and birds outside of the National Park boundaries. These migration patterns facilitate the potential transfer of ticks, presenting the opportunity for exchange of diverse tick species between the domestic, wild animal, and even human populations [1]. Therefore, the area is considered to be one of the hotspots for tick species, although the diversity of the tick species inhabiting the local area is scarce [1,10].

Studies aiming at quantifying and identifying tick species in Tanzania are still limited. The published studies related to tick species composition in Tanzania were conducted in Ngorongoro district [9], Iringa region, Maswa district [7], Mara region [8], Singida region, Mbeya region [6], where *Hyalomma rufipes, Rhipicephalus microplus,* and *Rhipicephalus evertsi* were reported. However, their information was restricted on the morphological characters only. The current study provides molecular diversity information on tick species and generates the mitochondrion 16S rRNA gene sequence data bases for the hard tick species infecting cattle, goats, and those collected from the environment in wildlife-livestock interface ecosystem of Mikumi National park, Tanzania. Molecular identification of *Ixodidae* ticks will provide valuable information to farmers and other stakeholders about different tick species present in the area [11]. Therefore, data from this field study can be useful for rational control strategies of ticks and tick-borne diseases in Tanzania [11,12].

## 2. Methods

### 2.1. Study Site

This study was conducted in the wards which lie at the border of Mikumi National Park, Morogoro region of Tanzania, namely, Doma, Melela, Kisaki, Tindiga, Kilangali, Ulaya, Mikumi, Ruhembe, Kidodi, and Kidatu (Figure 1). Mikumi National Park is located in the Morogoro region of Tanzania and lies between latitudes 7 °C and 10 °C south of the equator and between longitudes 36 °C and 37 °C East of Greenwich (Figure 1).The geography of the study area consists of plain land in most areas while some parts are covered by hills.

The area has typical tropical climate, with annual rainfall ranging from 600 to 1500 mm and average annual temperatures ranging from 20–30 °C. Rainfall distribution in the area is bimodal, with wet season from March to May. The dry seasons are experienced for six months (June, July, August, September, January, and February).

The area is the home to a pastoralist community for whom keeping livestock is their way of life. Poor animal husbandry and grazing practices put great pressure on land resources, which results in the need to continuously move large numbers of animals, especially cattle and goats, in search of pasture. This often brings livestock to share the pasture with wild animals in the wildlife-livestock interface ecosystem bordering the conserved area of Mikumi National Park. Nevertheless, significant human-animal conflicts are common as the wild animals straying from the reserve boundaries to the homesteads. This migration pattern facilitates the movement of potentially ticks across great distances, presenting the opportunity for the exchange of tick species between the domestic and wild animals.

Interaction between wild animals and livestock in the study area increase the likelihood of vectors and their pathogens parasitizing different vertebrate groups, resulting in pathogen spillover. In addition, ticks, among other vectors, may be facilitating the transmission of infectious pathogens among these groups of potential hosts.

### 2.2. Collection of Ticks

Ticks were collected from the body of domestic animals (632 ticks in cattle and 187 ticks in goats) as well as from the environment (28 free living, also termed questing ticks). A total of436 domestic animals were examined, including 260 cattle and 176 goats. All adult ticks from the animals were removed using forceps and placed in sterile plastic vials [13]. Questing (free living) ticks were collected from host resting areas and burrows, host routes and areas surrounding watering hole [4]. Collection methods for questing ticks included dragging of a flag, hand picking from vegetation and Carbon dioxide trapping using an improvised Carbon dioxide trap [4].

Following collection, the ticks were transported live to the laboratory in tubes plugged with cotton swabs. In the laboratory, the sampled ticks were washed with sterile water to remove excess environmental particulate contamination and then rinsed with 70% ethanol [11]. The washed ticks were then transferred to sterile vials and stored at −20 °C until processing for identification [4].

### 2.3. Tick Identification

#### 2.3.1. Morphological Identification of Ticks

Identification of ticks was done using morphological characteristics based on modified procedures as described by Walker et al., 2003 [5]. Sampled ticks were thawed at room temperature and rinsed once again with 70% ethanol. They were then mounted on slides and examined using a stereo microscope. Identification of the ticks was by genus using appropriate identification keys [5]. The morphologically identified ticks from all surveyed wards were pooled together into their respective genus.

#### 2.3.2. Molecular Identification of Ticks

For each genus of tick identified morphologically (*Rhipicephalus* and *Hyalomma*), 10 representative individuals were randomly selected for molecular analysis to assess the species diversity of the ticks in the wildlife-livestock interface of the ecosystem that lie at the border of Mikumi National Park and not for each specific ward.

##### DNA Extraction

The ticks were mechanically crushed in liquid nitrogen using a mortar and pestle followed by addition of 1 mL lysis buffer (NaCl 0.1 M, Tris-HCl 0.21 M, pH8 EDTA 0.05 M, SDS 0.5%). Enzymatic digestion of the hard tick protein cuticle was performed using the proteinase K. DNA extraction was carried out using phenol-chloroform extraction method [14]. The DNA was then precipitated with absolute ethanol before re-suspended in 200 µL of 1 × TE buffer (Tris 10 mM, EDTA 1 mM, pH8).

##### DNA Amplification

Specific primer setof 16S+1 (5′-CTGCTCAATGATTTTTTAAATTGCTGTGG-3′) and 16S-1 (5′-CCGGTCTGAACTCAGATCAAGT-3′) were used to target the mitochondrial 16S rRNA gene of ticks [15]. The PCR reactions were conducted in a final volume of 20μL; 10μLof PCR Master Mix, 6μLof nuclease free water, 1μLof 10 μmol/L of each primer, and 2μLof DNA template. The PCR condition for the 16S rRNA gene amplification was: initial denaturation 95 °C for 5 min; followed by 10 cycles of 92 °C for 1 min, 48 °C for 1 min and 72 °C for 30 s; 32 cycles of 92 °C for 1 min, 54 °C for 35 s, 72 °C for 30 s, followed by final extension of 72 °C for 7 min [16].

##### Agarose Gel Analysis and Sequencing of the 16S rRNA Amplicons

PCR products were visualized in 1.4% agarose gel(CSL-AG500, Cleaver Scientific Ltd., Rugby, UK) stained with EZ-vision^®^Bluelight DNA Dye (Bio-Rad Laboratories, Irvine, CA, USA). A100bp DNA ladder was used asa standard marker. PCR products were then purified using Qiagen PCR Purification Kit (Qiagen, Hilden, Germany) according to manufacturer’s protocol. Fragments were sequenced at Inqaba South Africa using the same forward and reverse primers as used to generate the PCR products. The labeled products were then cleaned with the ZR-96 DNA Sequencing Clean-up Kit (Zymoresearch, Irvine, CA, USA) (http://www.zymoresearch (accessed on 18 November 2019)). The cleaned products were injected on the Applied Biosystems ABI 3500XL Genetic Analyzer with a 50cm array using POP7 (Applied Biosystem, Foster city, CA, USA) (https://www.thermofisher.com (accessed on 18 November 2019)). Sequence chromatogram analysis was performed using Finch TV analysis software (Applied Biosystem, Foster city, CA, USA) (https://www.softpedia.com/get/Science-CAD/FinchTV.shtml (accessed on 18 November 2019)).

##### Sequences Editing, Blast Analysis, and Alignment

Sequences of the 16S rRNA gene from the present study were compared with the available data on GenBank using Blast on the NCBI website (https://blast.ncbi.nlm.nih.gov/Blast.cgi (accessed on 18 November 2019)) after trimming low-quality sequences at both ends and edited using BioEdit software in MEGA X (version 5.1). Representative mitochondrion 16S rRNA gene sequences of *Rhipicephalusmicroplus* and *R.evertsi* were downloaded from GenBank for analysis of *Rhipicephalus* species [17]. Similarly, mitochondrion 16S rRNA gene sequences for *Hyalommarufipes*, *H.marginatum*, *H.truncutum*, and *H. turanicum* for analysis of *Hyalomma* species were downloaded from GenBank [18] Multiple sequence alignments were then conducted using Clustal W in MEGA X software [19] The sequences acquired in this study have been deposited in the GenBank database with accession numbers MN961110 to MN961129.

##### Evolutionary Relationships of Taxa

The evolutionary history was inferred using the Unweighted Pair GroupMethod with Arithmetic mean (UPGMA) method [19]. The optimal tree with the sum of branch length = 0.54688856 was used. The percentage of replicate trees in which the associated taxa clustered together in the bootstrap test (1000 replicates) was used [19]. The tree was drawn to scale, with branch lengths in the same units as those of the evolutionary distances used to infer the phylogenetic tree. The evolutionary distances were computed using the Kimura 2-parameter method and were in the units of the number of base substitutions per site [19]. The analysis involved 26 nucleotide sequences. All ambiguous positions were removed for each sequence pair (pairwise deletion option) [19]. Evolutionary analyses were conducted in MEGA X [19].

## 3. Results

### 3.1. Data Acquisition and Morphological Identification of Ticks

The prevalence of tick-infestation in both cattle and goats were high (Table 1). From the animal’s body surface, the high tick infestation was found in udder, anal and ear in both cattle and goats. As shown in Table 2, there was variation in the mean tick intensity between cattle and goats.

Using the morphological characters as described by Walker et al., 2003 [5], ticks which belong into two genera namely; *Hyalomma* and *Rhipicephalus* were recorded in the present study (Table 3).

Both *Rhipicephalus* and *Hyalomma* tick spp. were collected in large amounts in cattle (Table 3). On the other hand, larger amounts of *Hyalomma* than *Rhipicephalus* tick spp. were collected in goats (Table 3). Very few amounts of adult ticks in both genera *Hyalomma* and *Rhipicephalus* ticks were collected from the environment in the present study (Table 3).

As indicated in Table 4, the mean abundance of ticks was 2.4 in cattle and 1.1 in goats, whereas the overall mean tick abundance was 1.8.

### 3.2. Molecular Identification and Classification of Ticks

DNA was isolated from 20 tick specimens selected randomly,10 from each genus of ticks initially identified by morphological characters. After partial amplification of the PCR products, all 20 samples were positive, and all were sequenced. The sequences of mitochondrion 16SrRNA fragments of 20 tick specimens used in this study were aligned and compared with the downloaded sequences from the GenBank (Table 5).

The lengths of the aligned sequences varied from 401 to 455 base pairs and the nucleotide components indicate that mitochondrion16S rRNA of these ticks is highly A-T rich with average nucleotide frequencies of Thymine (36.77%), Cytosine (9.51%), Adenine (39.71%), and Guanine (13.99%). Morphological identifications for ticks of the *Rhipicephalus* and *Hyalomma* genera were consistent with GenBank BLAST using sequences of the mitochondrion16S rRNA gene, with sequence identity (Table 5).

For the 16S rRNA gene sequences of *R. microplus* from the present study (Accession number MN961110, MN961111, MN961118, MN961122 and MN961125) the closest sequence was from an *R. microplus* isolate collected in Mozambique (GenBank: EU918187.1) and Thailand (GenBank:KT428016.1 and KC17O742.1) (Table 5).

For the 16S rRNA gene sequences of *R. evertsi* from the present study (Accession number MN961124), the closest sequence was from an *R. evertsi* isolate collected in South Africa (GenBank: KJ613642.1) (Table 5).

For the 16S rRNA gene sequences of *H. marginatum* (Accession number MN961114, MN961117, MN961123, MN961127), the closest sequences were from *H. marginatum* isolate collected from Algeria and Israel (GenBank: KP776645.1and KT391060.1), whereas *H. turanicum* (Accession number MN961115 and MN961126) from this study were closest to the sequence of *H. turanicum* isolate from Israel (GenBank:KT391063.1) (Table 5).

The sequences of *H. rufipes* (Accession number MN961111, MN961116, 961119, MN961120 and MN961129) from the present study were closest to the sequence of *H. rufipes* isolate from Egypt (GenBank: MK737649.1) and Hungary (GenBank: KU170517.1) (Table 5).

For the 16S rRNA gene sequences of *H. truncatum* (Accession number MN961113, MN961121, MN961128), the closest sequence was from a *H. truncatum* isolate collected from South Africa (GenBank: KU130478.1) (Table 5).

### 3.3. EvolutionaryRelationships of Taxa(Phylogenetic Analysis)

Based on the phylogenetic analysis of themitochondrion16S rRNA gene, two species have been reported in the *Rhipicephalus* genus, namely *R. microplus* and *R. evertsi* (Figure 2). All *R. microplus* and *R. evertsi* sequences generated in the present study clustered into one clade (B), together with other *R. mmicroplus* and *R. evertsi* isolates from other areas (Figure 2). For *Hyalomma* spp., sequences from the present study were selected for phylogenetic analysis and four species of *H. rufipes*, *H. truncatum*, *H. marginatum*, and *H. mturanicum* were recognized. Likewise, all the sequences of the *Hyalomma* spp. from the present study and GenBank formed another single clade (A) (Figure 2). However, *H. turanicum* (MN961115 and MN961126) formed a separate group from the *H. rufipes*, *H. truncatum*, and *H. marginatum* within the same clades. *H. rufipes* and *H. marginatum* collected during the present study clustered together and formed a complex group (Figure 2).

## 4. Discussion

Most of the previous studies from Tanzania were only based on morphological identification and most of them identified tick samples at the genus level [7,8]. There is therefore a need for molecular identification in order to generate genetic data bases for tick species and develop better control measures for ticks and tick borne pathogens [20,21].Two tick genera (*Rhipicephalus* and *Hyalomma*) were identified based on morphological characters. The two genera of hard ticks identified in this study, have also been reported by other researchers in some parts of Tanzania [6,7,8]. The presence of the tick spp. in the present study, similar to studies conducted in other sites in Tanzania, may be associated with unrestricted cattle movement from one area to another and cattle trade [22,23], which is a common phenomenon in Tanzania.

It is evident from the results in the present study that the mean tick intensity and the mean tick abundance differed between the two animal hosts (cattle and goats), which concur with previous studies [20,21]. The variations in infestation intensity and abundance could be due to a matter of feeding behavior differences between cattle and goats, as goats are browsers [21]. The observed high tick intensity in cattle as compared to goats may also be linked with the body surface area, host genetics, and small number of goats in our study [20]. However, limited information is available about ticks’ prevalence in small ruminants in Tanzania.

Most of the ticks in this study infested the sites with shorter hair and thinner skin [22]. The high tick infestations on these sites could be ascribed to the fact that ticks prefer warm, moist, and hidden sites with a good vascular supply and thin skin [23].

Six species have been confirmed based on mitochondrion16S rRNA analysis, including *R. microplus*, *R. evertsi*, *H. rufipes*, *H. truncatum*, *H. marginatum*, and *H. turanicum*. The evolutionary tree generated in the present study recorded several clusters of mitochondrial16S rRNA gene sequences, indicating divergence of gene sequences of hard ticks present in this wildlife-livestock interface ecosystem. Clusters of similar sequences represent species/subspecies clearly separated from other clusters (species/subspecies).

*R. microplus* was one among the tick species recorded in the present study. The finding of the *R. microplus* in the study region is in accordance with those of a local study conducted on Singida and Mbeya region of Tanzania [6], where the authors recorded high prevalence of *R. microplus*. The rapid expansion of *R. microplus* Tanzania is likely attributable to the shorter life-cycle and higher egg production capacity [24]. This phenomenon has recently been reported in several other African countries, including South Africa and Ivory Coast [25,26,27]. Moreover, the ability of *R. microplus* to develop resistance to most available acaricides might also have favored its expansion at the expense of more susceptible species [28]. According to Walker et al., 2003 [5], this species is widespread in tropical and subtropical regions and is considered to be the most important tick infesting livestock in the world. The higher prevalence of this species in the study area is of great interest because it is known to be a good vector of highly pathogenic *Babesia bigemina* and *Babesia bovis*, causing bovine babesiosis [29]. In addition, this species in terms of control management is well-known to be resistant to numerous pyrethroid and organophosphate compounds [30]. This species is also a vector of *Anaplasma marginale*, which causes anaplasmosis in cattle [31].

Another species belonging to the Rhipicephalus genus found in the present study was Rhipicephalus evertsi. This species has been reported with 33.8% prevalence in Mbeya region, Tanzania [6], to be prevalent in wild animals [7], and 10.9% prevalence in the Somali Regional State, Ethiopia [32]. According to Walker et al., 2003 [5], *R. evertsi* prefer the tropical geographical region in sub-Saharan Africa. Its distribution includes desert, steppe, savanna, and temperate climatic regions [5]. *R. evertsi* is of veterinary importance since it transmits the *Babesia caballi* and *Theileria equi* to horses, both causing forms of equine piroplasmosis [5]. Furthermore, this tick transmits the bacterium *Anaplasma marginale* to cattle causing bovine anaplasmosis. The saliva of female ticks contains a toxin that causes paralysis, particularly in lambs, but it may also affect calves and adult sheep [29].

*Hyalomma rufipes* was also recorded in the present study. This observation is in agreement with a previous report by Kerario et al., 2017 [6], with the highest prevalence (13.5% and 35.6%) in Singida and Mbeya regions, Tanzania respectively. However, this observation does not concur with Kwak et al., 2014 [7],who assessed Ixodidae tick infestation in Iringa and Maswa districts of Tanzania. In their study, this species was not observed. The observation of *H. rufipes* in the present study area is in agreement with Walker et al., 2003 [5], who assessed ticks of domestic animals in Africa. According to Walker et al., 2003 [5], *H. rufipes* is widely distributed in much of Africa and has been recorded from every climatic region from desert to rain forest. The presence of this species in the study area of Tanzania is of great importance, as it is known to be the most important vector in southern Africa of the virus causing Crimean-Congo hemorrhagic fever in human [29]. Furthermore, *H. rufipes* transmits *A. marginale,* a causative agent of anaplasmosis in cattle [33,34]. The feeding of adults on cattle causes large lesions at the attachment sites, leading to the formation of severe abscesses [3].

*Hyalomma marginatum* is another tick species that was recorded in the present study. According to Walker et al., 2003 [5] and Perveen et al., 2021 [35], *H. marginatum* occurs in areas with the humid Mediterranean climate of northern Africa and southern Europe and of steppe climate. The findings of Walker et al., 2003 [5] are not in agreement with our observation in the present study conducted in Morogoro region, which has hot and a moderate wet climate. The results of the current study could probably be due to changes in the tick populations since that time. There have been many changes since 2003 in the conditions required by the various tick species, including decrease of wild hosts, increase in the human population, increase in livestock and other domestic animals, the role of migratory birds in ticks transfer, changes in vegetation, changes in climate, changes in agricultural activities, increase in tick control regimes and its effectiveness, which together may well have had a great influence on the tick population [3]. The observation of this species in the study area is of great interest because it transmits the protozoa *Babesia caballi* causing babesiosis in horses, and it is known to transmit *Theileria annulata* causing theilerosis [3]. It is also responsible for transmission to humans of the virus causing Crimean-Congo haemorrhagic fever [29,35]. Furthermore, it can cause serious damage to cattle hides because of its long mouthparts. Notably, these ticks preferentially feed on the udder and teats of cattle and may cause serious problems in the suckling of calves [34].

*Hyalomma truncatum* is another tick species that was recorded in the present study. This observation is agreeable with Walker et al., 2003 [5] who assessed ticks of domestic animals in Africa. In their study, this species was observed to be endemic to the tropical region and thus is generally restricted to areas south of the Sahara, although it has been recorded from northern Sudan and from Egypt [35]. The observation of this species in the present study area is of great veterinary importance as certain strains of *Hyalomm truncatum* have a toxin in their saliva that causes the skin diseases known as sweating sickness in cattle. The long mouth-parts cause tissue damage and secondary bacterial infections may lead to infected abscesses [5].

The last and most interesting tick species found in the present study, *Hyalommat ranicum*, had not been reported from the study region before. According to Walker et al., 2003 [5], *H. turanicum* is distributed in areas with steppe and desert climate, and in Africa it occurs mainly in the central and western arid regions of southern Africa. The results of the current study could probably be due to a change in the tick populations since that time, as there have been enormous changes since 2003 in the conditions required by the various tick species, same conditions as discussed previously for *H. marginatum* apply to both species [3]. The presence of this species in the study area of Tanzania is of great importance as it is considered to be vector of the virus causing Crimean-Congo hemorrhagic fever in humans [29,35], although *H. turanicum* is not known to be a main vector of pathogens causing disease in domestic animals.

## 5. Conclusions

Present study provides genetic characterization of the mitochondrion16SrRNAgenes diversity of the hard ticks collected from the wildlife-livestock interface at Mikumi National Park, Morogoro region, Tanzania. The tick species infesting domestic animals (cattle and goats) in the study area were *Rhipicephalus microplus*, *Rhipicephalus evertsi*, *Hyalomma rufipes*, and *Hyalomma truncatum*. *Hyalomma marginatum* and *Hyalomma turanicum* were also found, whereby these two species had not been reported from the study region before. The presence of different groups of tick species and the observation of the species which had not been reported from the study region before, reflect the possible biological diversity of hard ticks present in this wildlife-livestock interface ecosystem. Therefore, further work is required to investigate species of hard tick present in the study region over a larger scale.

## Figures and Tables

**Figure 1 vetsci-08-00036-f001:**
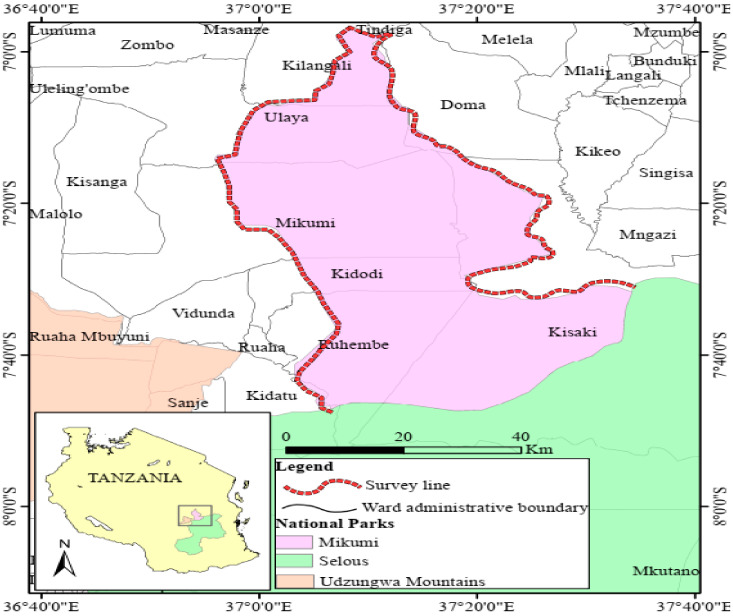
Map showing wildlife-livestock interface ecosystem of Mikumi National Park. Data source; (National Bureau of Statistics (Geograhic Information System) data base; https://www.nbs.go.tz/index.php/en/) (accessed on 18 November 2019).

**Figure 2 vetsci-08-00036-f002:**
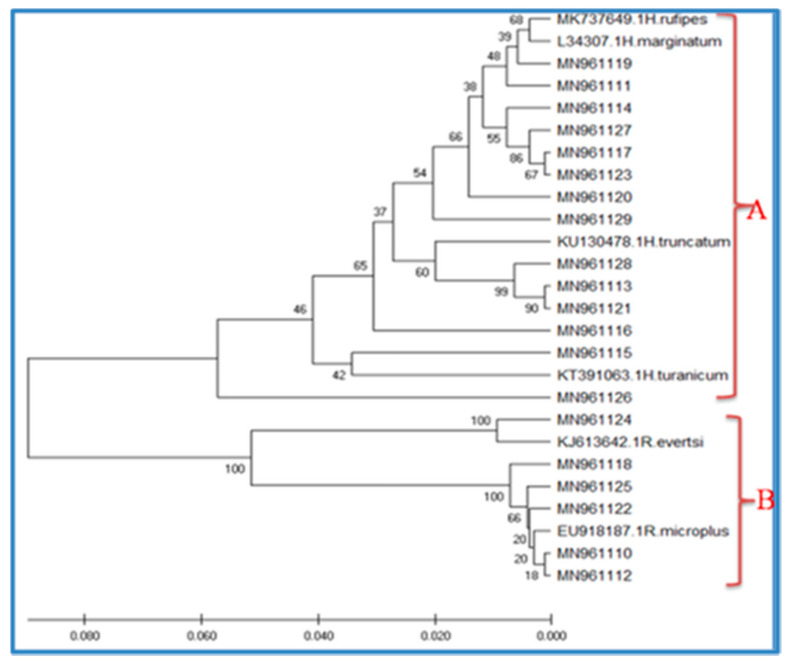
Overall phylogenetic tree (inferred using the Unweighted Pair Group Method with Arithmetic mean (UPGMA)) for tick species based on the 16S rRNA gene, including sequences obtained in the present study and representative sequences of the known species from GenBank (indicated with a species name). A and B in the phylogenetic tree are the clades.

**Table 1 vetsci-08-00036-t001:** Prevalence of ticks in cattle and goats ^a^.

Animals	Type		Total
	Cattle N (%)	Goats N (%)	
Examined	260	176	436
Infested with ticks	134 (51.5)	78 (44.3)	212 (48.6)

^a^ Number of infested animals/Number of examined animal× 100.

**Table 2 vetsci-08-00036-t002:** Mean ticks intensity in cattle and goats ^b^.

Category	Cattle	Goats	Total
Animals infested	134	78	212
Ticks collected	632	187	819
Mean tick intensity	4.7	2.4	3.9

^b^ Number of collected ticks/Number of infested animals.

**Table 3 vetsci-08-00036-t003:** Number (N) of ticks collected from cattle, goats, and environment (free) ^c^.

Tick Genus	Source			Total
	Cattle N (%)	Goats N (%)	Free N (%)	
*Rhipicephalus*	378 (59.8)	36 (19.3)	16 (57.1)	430 (51)
*Hyalomma*	254 (40.2)	151 (80.7)	12 (42.9)	417 (49)
Total	632 (100)	187 (100)	28 (100)	847 (100)

^c^ Number of tick spp. from (cattle, goats, environment)/Total tick spp. from (cattle, goats, environment) × 100.

**Table 4 vetsci-08-00036-t004:** Mean ticks abundance in cattle and goats ^d^.

Category	Cattle	Goats	Total
Animal analyzed	260	176	436
Ticks collected	632	187	819
Mean abundance	2.4	1.1	1.8

^d^ Nunber of ticks collected/Number of analyzed animals.

**Table 5 vetsci-08-00036-t005:** Identity of tick species and percentage similarity value with the reference sequences.

Sample Accession	Source	Site/Ward	GeneBank Reference	Identity %	Species
MN961110	Cattle	Doma	EU918187.1	100	*Rhipicephalusmicroplus*
MN961111	Environment	Mikumi	KU170517.1	98.76	*Hyalomma refipes*
MN961112	Cattle	Mangae	KT428016.1	100	*Rhipicephalusmicroplus*
MN961113	Cattle	Kidodi	KU13478.1	97.15	*Hyalomma truncatun*
MN961114	Cattle	Mkata	KP776645.1	98.99	*Hyalomma marginatum*
MN961115	Cattle	Mikumi	KT391063.1	93.98	*Hyalomma turanicum*
MN961116	Cattle	Doma	MK737649.1	94.70	*Hyalomma rufipes*
MN961117	Cattle	Ulaya	KP776645.1	98.74	*Hyalomma marginatum*
MN961118	Cattle	Melela	KT428016.1	99.75	*Rhipicephalusmicroplus*
MN961119	Environment	Kilangali	KU170517.1	98.99	*Hyalomma rufipes*
MN961120	Goat	Doma	KU170517	99.97	*Hyalomma rufipes*
MN961121	Cattle	Ruhembe	KU130478.1	96.63	*Hyalomma truncatum*
MN961122	Environment	Mangae	KT428016.1	99.26	*Rhipicephalusmicroplus*
MN961123	Goat	Mikumi	KT391060.1	98.50	*Hyalomma marginatum*
MN961124	Cattle	Mkata	KJ613642.1	98.42	*Rhipicephalus evertsi*
MN961125	Goat	Tindiga	KC170742.1	99.50	*Rhipicephalusmicroplus*
MN961126	Cattle	Melela	KP776645	91.54	*Hyalomma turanicum*
MN961127	Cattle	Kisaki	KP776645.1	98.50	*Hyalomma marginatum*
MN961128	Goat	Mangae	KU130478.1	96.37	*Hyalomma truncatum*
MN961129	Cattle	Kidatu	MK737650.1	97.47	*Hyalomma rufipes*

## Data Availability

The data supporting the conclusions of this article are included within the article. The sequences generated in the present study were submitted to GenBank under the accession numbers MN961110 to MN961129.

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
