# Peer review of "Molecular Diversity of Hard Tick Species from Selected Areas of a Wildlife-Livestock Interface Ecosystem at Mikumi National Park, Morogoro Region, Tanzania"

_vetsci, 2021, doi:10.3390/vetsci8030036_

Round 1

Reviewer 1 Report

The present study aims to assess the diversity of hard tick species present in selected areas of a region considered to be a wildlife-livestock interface by means of both morphological and molecular identification. According to the authors, it represents the first study including molecular identification of ticks in Tanzania, which deserves interest.

My main concerns with this manuscript include:

  • I’m afraid that it suffers of over-concluding derived of a small sample size and low representativeness preventing concluding on the most common tick species infesting cattle and goats in the study area. Only 20 ticks identified in a study area with an unknown surface (not provided). According to Figure 1 and its scale, the study area covers lineal distances over 500 km, which would result in a total surface ranging at least 250000-500000 km2. However, taking into account that Tanzania has a total surface of 947300 km2(wikipedia), this is unfeasible and there is probably a mistake in the scale provided in Figure 1. In any case, I consider 20 ticks a small number. In addition, the spatial distribution of the origin of the identified ticks is not mentioned.
  • The “wildlife-livestock interface” concept is mentioned in the title and repeated over and over again along the text. However, no wild animals have been sampled nor information regarding the main wildlife species present in the study area is included.
  • Morphological identification to the level species has not been apparently attempted with a bigger number of ticks, in addition to those molecularly identified. To be fair, I do not know if this is feasible, despite other studies providing tick identification results to the level species in Tanzania (supposedly through morphological identification) are cited.
  • Lastly, English composition is, to my opinion, quite unsatisfactory and needs editing and be shortened as some ideas are repeated over and over again unnecessarily instead of merging assessments that apply to several of the identified tick species, for example.

The authors can find some additional comments in the revised manuscript file. 

Author Response

Open Review 1

English language and style

(x) Extensive editing of English language and style required
( ) Moderate English changes required
( ) English language and style are fine/minor spell check required
( ) I don't feel qualified to judge about the English language and style

Yes

Can be improved

Must be improved

Not applicable

Does the introduction provide sufficient background and include all relevant references?

( )

(x)

( )

( )

Is the research design appropriate?

( )

(x)

( )

( )

Are the methods adequately described?

( )

(x)

( )

( )

Are the results clearly presented?

( )

(x)

( )

( )

Are the conclusions supported by the results?

( )

( )

(x)

( )

Comments and Suggestions for Authors

The present study aims to assess the diversity of hard tick species present in selected areas of a region considered to be a wildlife-livestock interface by means of both morphological and molecular identification. According to the authors, it represents the first study including molecular identification of ticks in Tanzania, which deserves interest.

My main concerns with this manuscript include:

  • I’m afraid that it suffers of over-concluding derived of a small sample size and low representativeness preventing concluding on the most common tick species infesting cattle and goats in the study area. Only 20 ticks identified in a study area with an unknown surface (not provided).

Response: Thank you very much for your valuable time and comments on our manuscript. We are highly appreciating your point of view.  Because the aim of the study was to have the existing molecular diversity of ticks in an interface ecosystem, ticks were collected from cattle, goats and from the environment before morphological identification into the genus level.  The morphologically identified ticks were pooled together into their respective genus. For each genus pool of ticks identified morphologically, ten representative individuals were randomly selected for molecular analysis to have the existing genetic diversity of the ticks in the ecosystem that lie at the border of Mikumi National Park and not for the ticks from the wild animals. Therefore, we sequenced 10 ticks from each genus making a total of 20 ticks from two genera (10 Rhipicephalus and 10 Hyalomma). In our opinion, this approach is valid since the ecosystem at the border of Mikumi National Park is interconnected, for example by herding of livestock and movement of human and wildlife across ward borders. We have rephrased the sentence “the most common tick species…..’ as per the instruction given’ (see page 9).

  • According to Figure 1 and its scale, the study area covers lineal distances over 500 km, which would result in a total surface ranging at least 250000-500000 km2. However, taking into account that Tanzania has a total surface of 947300 km2(wikipedia), this is unfeasible and there is probably a mistake in the scale provided in Figure 1.

Response: Many thanks for this very useful observation and comment. It is very true that the scale was unfeasible; there was a mistake in the scale provided. We have recognized and fixed the problem and now the scale is feasible (Page 3).

  • In any case, I consider 20 ticks a small number. In addition, the spatial distribution of the origin of the identified ticks is not mentioned.

Response: Many thanks for this comment. As mentioned in our previous response above, the morphologically identified ticks from all surveyed wards were pooled together into their respective genus. For each genus of tick identified morphologically, ten representative individuals were randomly selected for molecular analysis to have the existing genetic diversity of the ticks in the human-wildlife-livestock interface of the ecosystem that lie at the border of Mikumi National Park and not for each specific ward. Therefore, we sequenced 10 ticks from each genus making a total of 20 ticks from the two genera (10Rhipicephalus and 10 Hyalomma). In our opinion, this approach is valid since the ecosystem at the border of Mikumi National Park is interconnected, for example by herding of livestock and movement of human and wildlifeacross ward borders.

  • The “wildlife-livestock interface” concept is mentioned in the title and repeated over and over again along the text. However, no wild animals have been sampled nor information regarding the main wildlife species present in the study area is included.

Response: Thank you for this comment. The aim of the study was to collect the ticks from cattle, goats and environment at the border lying ecosystem between livestock land and the National Park (wildlife-livestock interface) and not from the wild animals. We agreed with you and we have made revision in some repeated “ wildlife-livestock” words. See for example pages 1, 2 and 9.

  • Morphological identification to the level species has not been apparently attempted with a bigger number of ticks, in addition to those molecularly identified. To be fair, I do not know if this is feasible, despite other studies providing tick identification results to the level species in Tanzania (supposedly through morphological identification) are cited.

  • Response: Thank you for this comment. The central goal of the study was to have molecular diversity of the tick species and to generate the 16S rRNAdata basefor the hard ticks collected in the named ecosystem and Tanzania in general.

  • Lastly, English composition is, to my opinion, quite unsatisfactory and needs editing and be shortened as some ideas are repeated over and over again unnecessarily instead of merging assessments that apply to several of the identified tick species, for example.

Response: Thank you for this opinion. Editing have been done, some repeated ideas have been removed.

The authors can find some additional comments in the revised manuscript file. 

Response: Many thanks for the additional comments in the revised manuscript file. We found very useful and relevant comments, opinions and information. We have used them accordingly in making the revision of the manuscript.

Reviewer 2 Report

The manuscript Diversity of hard ticks (Ixodidae) from select areas of a wildlife-livestock interface ecosystem at Mikumi National Park, Morogoro region, Tanzania by Damian and others report tick species and genetic population in a region of Tanzania using morphological and molecular tools. In my opinion, this manuscript requires a major revision before it can be considered for publication. Also, in light of the limited sample size used for molecular analysis (20) and lack of robustness in the sampling method, this manuscript must be revised as a short communication.
One detail, the manuscript does not have lines numbered, which makes it very difficult to review.

Major issues:
Tittle
Need no to be reformulated since the main component of the work, which is the novelty, is not presented in the title. The molecular (genetic) study of tick populations.

The abstract is poorly written. Too many repetitive words, missing oxford comma, lack of clear objective.
I don´t think the first paragraph is necessary in the abstract.
“were identified using morphological characters initially”
Remove the word initially
Why at least? This is confusing.. I understand only after reading the method section. So it would be better if the abstract was self-explained.
Abstracts do not have any information about how many ticks were collected…

Introduction:
“However, their information is restricted to morphology characters”
Not clear! Authors must inform in the introduction the following:
Was published work in the SAME area? Yes or no?
If there are, which tick species they found?
IF in the same area, why this study is important. Molecular identification alone does not justify a new study if the species diversity was already studied in the same region. Did the authors obtain a larger sample?

Methods
“Questing (free living) ticks were collected from host resting areas and burrows, host routes and areas surrounding watering hole”
There is no information about the sampling method. It was flagging?? Please add details about questing ticks sampling method.
More information is necessary regarding animal sampling methods. How many animals, sample size calculation, sampling approach (how many animals per farm, how many farms/owners). How they collected the ticks from animals? How many ticks per animal were collected (all?). Section 2.2 needs an expansion with complete details.

Results
“Total of 436 domestic animals were examined, including 260 cattle and 176 goats (Table1)”
I think this information must be given at the methods.
Parasitological parameters are required in this study (infestation intensity, abundance, and mean crowding).
What about the development stage? There is no information whatsoever about the development stage of sampled ticks (nymphs, adults, larvae).

Minor issues: 

Introduction
Therefore, the area is considered to be one of the hotspot points for tick species, although the genetic diversity of the tick species inhabiting the local area is still unknown
Need a citation.

“in Ngorongoro [9], Iringa, Maswa [7], Mara [8], Singida, Mbeya”
This information means nothing to international readership. Authors must explain the names are related to what. Cities? Areas of the park??

Methods
“Doma, Melela, Kisaki, Tindiga, Kilangali, Ulaya, Mikumi, Ruhembe, Kidodi and Kidatu”
Again, these names are referring to what?

Results
“A total of 847 ticks were collected from the animals (infesting ticks) as well as from the environment (free ticks)”
Poor writing, the readers need to calculate to find how much questing tick did you sampled. Be clear. XX ticks were sampled from animals and yyy ticks were sampled from the environment.

Discussion
“We conducted a survey on hard ticks collected from domestic animals (cattle and goats) and environment (free-living) in a wildlife-livestock interface of Mikumi National Park, Morogoro region, Tanzania.”
This is methods, remove it from here.

“Only adult ticks were identified because immature ticks (larva and nymph) lack important morphological features required for identification”
This is not true. There is an available identification key for nymphs.

“There is therefore a need for molecular identification in order to develop better control measures for ticks and tick borne pathogens”
Need a citation. I am not sure about this…

Author Response

Open Review 2

English language and style

( ) Extensive editing of English language and style required
(x) Moderate English changes required
( ) English language and style are fine/minor spell check required
( ) I don't feel qualified to judge about the English language and style

Yes

Can be improved

Must be improved

Not applicable

Does the introduction provide sufficient background and include all relevant references?

( )

( )

(x)

( )

Is the research design appropriate?

( )

( )

(x)

( )

Are the methods adequately described?

( )

( )

(x)

( )

Are the results clearly presented?

( )

( )

(x)

( )

Are the conclusions supported by the results?

( )

( )

(x)

( )

Comments and Suggestions for Authors

The manuscript Diversity of hard ticks (Ixodidae) from select areas of a wildlife-livestock interface ecosystem at Mikumi National Park, Morogoro region, Tanzania by Damian and others report tick species and genetic population in a region of Tanzania using morphological and molecular tools. In my opinion, this manuscript requires a major revision before it can be considered for publication. Also, in light of the limited sample size used for molecular analysis (20) and lack of robustness in the sampling method, this manuscript must be revised as a short communication.
Response: Thank you very much for your valuable time and comments on our manuscript. We are highly appreciating your point of view. The morphologically identified ticks were pooled together into their respective genus. For each genus pool of ticks identified morphologically, ten representative individuals were randomly selected for molecular analysis to have the existing genetic diversity of the ticks in the ecosystem that lie at the border of Mikumi National Park and not for the ticks from the wild animals. Therefore, we sequenced 10 ticks from each genus making a total of 20 ticks from two genera (10 Rhipicephalus and 10 Hyalomma). In our opinion, this approach is valid since the ecosystem at the border of Mikumi National Park is interconnected, for example by herding of livestock and movement of human and wildlife across ward borders.   

Major issues:
Tittle
Need no to be reformulated since the main component of the work, which is the novelty, is not presented in the title. The molecular (genetic) study of tick populations.

Response: Thank you for this comment. We have reformulated the title as per opinion given (Page 1).

The abstract is poorly written. Too many repetitive words, missing oxford comma, lack of clear objective.

Response: Thank you for this comment. The comments have been addressed and the abstract now have been revised accordingly (Page 1).

I don´t think the first paragraph is necessary in the abstract.  “were identified using morphological characters initially”

Response: Thank you for this comment. The paragraph has been removed as per a suggestion given (Page 1)

Remove the word initially

Response: Thank you for this comment. The word initially has been removed (Page 1).

Why at least? This is confusing. I understand only after reading the method section. So it would be better if the abstract was self-explained.

Response: Thank you for this useful suggestion.  The sentence has been rephrased (page 1)

Abstracts do not have any information about how many ticks were collected…

Response: Thank you for this comment. Revision for the abstract has been done and the number of ticks collected now appears in the abstract (page 1).

Introduction:
“However, their information is restricted to morphology characters”
Not clear! Authors must inform in the introduction the following:
Was published work in the SAME area? Yes or no?
If there are, which tick species they found?
IF in the same area, why this study is important. Molecular identification alone does not justify a new study if the species diversity was already studied in the same region. Did the authors obtain a larger sample?

Response: Thank you for this useful comment in the form of questions. Revision for this section has been done, the introduction now is clear and the questions raised have been addressed in the introduction part as per opinion given (Page 2).

Methods
“Questing (free living) ticks were collected from host resting areas and burrows, host routes and areas surrounding watering hole”
There is no information about the sampling method. It was flagging?? Please add details about questing ticks sampling method.
More information is necessary regarding animal sampling methods. How many animals, sample size calculation, sampling approach (how many animals per farm, how many farms/owners).How they collected the ticks from animals? How many ticks per animal were collected (all?). Section 2.2 needs an expansion with complete details.

Response: Thank you for this very useful comment. Revision for this section has been done. The details  about how the free living ticks were collected, more information about regarding animal sampling have been stated clearly ( Page 4).

Results
“Total of 436 domestic animals were examined, including 260 cattle and 176 goats (Table1)”
I think this information must be given at the methods.

Response: Thank you for this comment. Revisions for these sections have been done and the stated information is now in the methods part (Page 5).

Parasitological parameters are required in this study (infestation intensity, abundance, and mean crowding).

Response: Thank you for this comment. Revision has been done and the parasitological parameters are now inserted in the manuscript as per suggestion given (Page 6).

What about the development stage? There is no information whatsoever about the development stage of sampled ticks (nymphs, adults, larvae).

Response: Many thanks for this comment. We agree with this comment; however the set-up of the present study was to collect the adult ticks and not the lava or nymph.

Minor issues: 

Introduction
Therefore, the area is considered to be one of the hotspot points for tick species, although the genetic diversity of the tick species inhabiting the local area is still unknown
Need a citation.

Response: Thank you for this opinion. We have put the citation in this sentence (Page 2).

“inNgorongoro [9], Iringa, Maswa [7], Mara [8], Singida, Mbeya”
This information means nothing to international readership. Authors must explain the names are related to what. Cities? Areas of the park?

Response: Many thanks for this comment. We have rephrased this sentence and it now hopefully looks clearer (page 2).

Methods
“Doma, Melela, Kisaki, Tindiga, Kilangali, Ulaya, Mikumi, Ruhembe, Kidodi and Kidatu”
Again, these names are referring to what?

Response: Many thanks once again for this comment. These refer to the wards. We have rephrased this sentence and hope it now is clear what we mean (Page 3).

Results
“A total of 847 ticks were collected from the animals (infesting ticks) as well as from the environment (free ticks)”
Poor writing, the readers need to calculate to find how much questing tick did you sampled. Be clear. XX ticks were sampled from animals and yyy ticks were sampled from the environment.

Response: Thank you for this comment. We agree on this, now we have specified the number ticks collected from the environment and the ticks collected from each study animal (Page 6).

Discussion
“We conducted a survey on hard ticks collected from domestic animals (cattle and goats) and environment (free-living) in a wildlife-livestock interface of Mikumi National Park, Morogoro region, Tanzania.”
This is methods, remove it from here.

Response: Thank you for this comment. For sure we agree with this comment that this must be in the methodology part, we have removed it from this section (Page 9).

“Only adult ticks were identified because immature ticks (larva and nymph) lack important morphological features required for identification”
This is not true. There is an available identification key for nymphs.

Response: Thank you for this comment. We agree with this comment, although our study set up was to collect the adult ticks and not the immature ticks. But we have rephrased the sentenceto be clearer on this point (page 9).

“There is therefore a need for molecular identification in order to develop better control measures for ticks and tick borne pathogens”
Need a citation. 

Response: The citation has been inserted as instructed (page 9).

Reviewer 3 Report

The article by Damian et al. utilize mitochondrion 16S rRNA gene to identify morphologically identified ticks from Mikumi National Park, Tanzania.

I have following comments:

  • Methods: In collection of ticks section, Ticks was collected from animals and areas surrounding using what? Please state, respectively.
  • Evolutionary relationship of taxa section, 100 replicates (bootstrap test) is low and not reliable, the author should repeat the tree with 1000 replicates
  • Was the phylogenetic tree constructed by the Neigbhor-joining (NJ) method? Please state the method used.
  • Table 3, remove column “Area” Mozambique to Egypt. The information is already provided in the text where it is appropriate. Providing such information in the Table is misleading and can be confusing to be sample source country.
  • Discussion section, Fourth paragraph “Of the six species that were …….. micropluswas the most abundant in the study area. Our finding that ……” These statements are wrong and speculative, how did the authors came with this conclusion? when they only identified 10 specimens/genus to species level. The authors should re-write the statement and stick to their data.
  • “The higher prevalence of the species in the study area” Again, No data for this statement in the study.
  • Sixth paragraph “Hyalomma rufipesis the second sp that was recorded in great abundance in the study area” No data showing this statement in their study.
  • This difference could be due to temperature, rainfall pattern….. in the areas” This suggestion was based on what? Any citation for it?

Generally, the observed results are poorly discussed. Most claims in the discussion is misleading and no data for most statements in their study. The authors should apply some caution with their wide/generalized statements. Therefore, I strongly recommend a major revision of the manuscript, especially improving the discussion section The author should carefully discuss their data and avoid misleading statements.

Some scientific names are not italicized. Check through and correct appropriately.

Author Response

Open Review 3

English language and style

( ) Extensive editing of English language and style required
( ) Moderate English changes required
(x) English language and style are fine/minor spell check required
( ) I don't feel qualified to judge about the English language and style

Yes

Can be improved

Must be improved

Not applicable

Does the introduction provide sufficient background and include all relevant references?

(x)

( )

( )

( )

Is the research design appropriate?

( )

(x)

( )

( )

Are the methods adequately described?

( )

(x)

( )

( )

Are the results clearly presented?

( )

(x)

( )

( )

Are the conclusions supported by the results?

( )

( )

(x)

( )

Comments and Suggestions for Authors

The article by Damian et al. utilize mitochondrion 16S rRNA gene to identify morphologically identified ticks from Mikumi National Park, Tanzania.

I have following comments:

  • Methods: In collection of ticks section, Ticks was collected from animals and areas surrounding using what? Please state, respectively.

Response: Thank you for this very useful comment. We have made correction as per suggestion; ticks collection method has now been stated more clearly (Page 4).

  • Evolutionary relationship of taxa section, 100 replicates (bootstrap test) is low and not reliable, the author should repeat the tree with 1000 replicates.

Response: Thank you very much for this comment. We agree your opinion, we have repeated the evolutionary relationship tree with 1000 replicates (bootstrap test), although we have obtained almost the same evolutionary tree(Page 9).

  • Was the phylogenetic tree constructed by the Neigbhor-joining (NJ) method? Please state the method used.

Response: Thank you for this very useful comment. We have now stated the method used to construct the phylogenetic tree (page 5).

  • Table 3, remove column “Area” Mozambique to Egypt. The information is already provided in the text where it is appropriate. Providing such information in the Table is misleading and can be confusing to be sample source country.

Response: Many thanks for this comment. We have removed column “Area” from table 3 (page 8).

  • Discussion section, Fourth paragraph “Of the six species that were …….. micropluswas the most abundant in the study area. Our finding that ……” These statements are wrong and speculative, how did the authors came with this conclusion? when they only identified 10 specimens/genus to species level. The authors should re-write the statement and stick to their data.

Response: We strongly agree this comment. We have rephrased the sentence as per instruction given (page 10).

  • “The higher prevalence of the species in the study area” Again, No data for this statement in the study.

Response: For sure we agree this comment. We have rephrased the sentence and now it sounds better (page 10).

  • Sixth paragraph “Hyalommarufipesis the second sp that was recorded in great abundance in the study area” No data showing this statement in their study.

Response:Thank you for this very useful comment. We have rephrased the sentence and now it is clearer and more understandable (10).

  • This difference could be due to temperature, rainfall pattern….. in the areas” This suggestion was based on what? Any citation for it?

Response: Thank you for this comment. We have removed this sentence (page11).

Generally, the observed results are poorly discussed. Most claims in the discussion is misleading and no data for most statements in their study. The authors should apply some caution with their wide/generalized statements. Therefore, I strongly recommend a major revision of the manuscript, especially improving the discussion section .The author should carefully discuss their data and avoid misleading statements. Some scientific names are not italicized. Check through and correct appropriately.

Response: Thank you for the valuable and good comments. We have addressed the comments and opinions and we think the incorporated responses make the discussion and other parts of the manuscript now look much better. All scientific names have been italicized now.

Round 2

Reviewer 1 Report

Dear Authors,

Congratulations for this improved version of your manuscript on the diversity of hard tick species from selected wildlife-livestock interface areas in Tanzania; you have successfully addressed most of my concerns.

However, I still consider that the novel results provided, despite deserving interest, could be suitably communicated and discussed in the form of a short communication, in a more straightforward way.

In addition, please pay attention to editing/typewriting mistakes such as lacking spaces between words in the text or proper English editing (too many to be detailed).

Best regards. 

Author Response

Comment: Congratulations for this improved version of your manuscript on the diversity of hard tick species from selected wildlife-livestock interface areas in Tanzania; you have successfully addressed most of my concerns.

Response: Many thanks to you for the positive and valuable comments and recommendations provided to our revised manuscript. We highly appreciate your time, efforts, hard working and point of view in improving and fine-tuning our manuscript for publication.

Comment: However, I still consider that the novel results provided, despite deserving interest, could be suitably communicated and discussed in the form of a short communication, in a more straightforward way.

 Response: Thank you for this comment: We are appreciating your point of view. Since the manuscript covers all necessary requirements for the article (Background, well organized methodology, results coverage is very wide and very detailed discussion and the manuscript contain 14 pages), as you have indicated the deserving interest of the manuscript we are requesting this manuscript to be accepted as full research article paper. As mentioned in our previous response that the wards where we collected the samples are interconnected and the livestock keeping is under free range (nomadic pastoralism) which allow the interaction between the animals across the wards. For our opinion, the random selection of 20 specimens, 10 from each genus of the ticks identified morphologically for molecular analysis can homogeneously represent the study site.

Comment: In addition, please pay attention to editing/typewriting mistakes such as lacking spaces between words in the text or proper English editing (too many to be detailed).

Response:  Thank you for the given attention.  We have addressed the mentioned problems to our level best and the manuscript now looks much better.

Reviewer 2 Report

The authors successively address all my comments and provide a detailed revision note. It is very rewarding when authors value the reviewer's work and correct the manuscript accordingly. I believe that this paper, although with some limitations previously mentioned, has new information from a poorly studied area and deserves to be published in the journal.  

Unfortunately, in my opinion, this manuscript must be accepted as a short communication. 

There are some problems with the lack of space between words, please check this during proof-reading. 

I also recommend the authors to include a recent review about ticks in Africa (although it was in North Africa, I think it would be important to include): https://doi.org/10.3390/insects12010083

There is one problem, the authors did not include the requested parasitological parameters: 

infestation intensity

abundance

mean crowding

Please add this data in a table. And add some discussion about it. If you need information on how to calculate this, you can check this paper: https://doi.org/10.1186/s13071-018-2917-2

There is a software that you can use for the calculations.  

Author Response

Comment: The authors successively address all my comments and provide a detailed revision note. It is very rewarding when authors value the reviewer's work and correct the manuscript accordingly. I believe that this paper, although with some limitations previously mentioned, has new information from a poorly studied area and deserves to be published in the journal. 

Response: Thank you very much for your comments on our revised manuscript, we are happy for your opinion that our manuscript can now be considered for publication. Indeed as you have noted our manuscript provides a holistic, balanced, and intensive recapitulation by considering the value of the livestock in the livelihoods and economies of peoples and countries, we are requesting our manuscript to be accepted as a full article paper.

Comment: Unfortunately, in my opinion, this manuscript must be accepted as a short communication. 

 Response: Thank you for this comment: We are appreciating your point of view. Since the manuscript covers all necessary requirements for the article (Background, well-organized methodology, results in coverage is a very wide and very detailed discussion and the manuscript contain 14 pages), as you have indicated the deserving interest of the manuscript (new information from a poorly studied area) we are requesting this manuscript to be accepted as full research article paper. As mentioned in our previous response that the wards where we collected the samples are interconnected and the livestock keeping is under free-range movements which make the interaction of animals. In our opinion, the random selection of 20 specimens, 10 from each genus of the ticks identified morphologically for molecular analysis can homogeneously represent the study site.

Comment: There are some problems with the lack of space between words, please check this during proof-reading. 

Response: Thank you for this observation. We have addressed the problems accordingly (pages 1 to 14).

Comment: I also recommend the authors to include a recent review about ticks in Africa (although it was in North Africa, I think it would be important to include): https://doi.org/10.3390/insects12010083

Response: Thank you very much for this recommendation, we are appreciating your valuable time and effort to improve our manuscript. We have passed through the recent review provided via the link with the title “Ticks and Tick-borne diseases of livestock in the Middle East and North Africa” Indeed it is very useful to review. We have cited the review and included it in the manuscript, please refer to the discussion part (page 11) and reference part (page 14).

Comment: There is one problem, the authors did not include the requested parasitological parameters: 

infestation intensity

abundance

mean crowding

Please add this data in a table. And add some discussion about it. If you need information on how to calculate this, you can check this paper: https://doi.org/10.1186/s13071-018-2917-2

There is a software that you can use for the calculations.  

Response:  Thank you very much for the very useful observation and comments, the data on the infestation intensity, abundance, and mean infestation have been added in the manuscript (page 6) and discussed (page 10). Thank you very much for the paper provided via the link, it was very useful and indeed helped us in addressing the provided comments.

Reviewer 3 Report

In the revised version of their manuscript, the authors have address the deficiencies. However, I have the following few comments.

  • A figure or table should be self-explanatory without referring to the main text of the article. Thus, the authors should add what A and B (clades?) are, in the figure 2 legend, for easy guide to the readers.
  • Also, the authors should state if the phylogenetic tree was inferred by using Maximum Likelihood method or Neighbour-joining method in the figure 2 legend and section 2.3.2.5 (Evolutionary relationships of taxa).
  • Overall, there some words that suppose to be separated in the text but joined. For instance, Total "of436", "mitochondrial16S" etc. The authors should carefully read through the manuscript and correct them and others. 

Author Response

Comment: In the revised version of their manuscript, the authors have address the deficiencies.

Response: Many thanks to you for the positive and valuable comments and recommendations provided to our revised manuscript. We highly appreciate your time, efforts, and point of view in improving and fine-tuning our manuscript for publication.

Comment: However, I have the following few comments.

A figure or table should be self-explanatory without referring to the main text of the article. Thus, the authors should add what A and B (clades?) are, in the figure 2 legend, for an easy guide to the readers.

Response:  Thank you very much for the comment. We have addressed the problem in the figure 2 legend for an easy guide to the leader (page 9).

Comment: Also, the authors should state if the phylogenetic tree was inferred by using Maximum Likelihood method or Neighbour-joining method in the figure 2 legend and section 2.3.2.5 (Evolutionary relationships of taxa).

Response:  Thank you very much for the comment.  The phylogenetic tree was inferred using Unweighted Pair GroupMethod with Arithmetic mean (UPGMA) method (page 5). We have also stated this in the figure legend (page 9).

Comment: Overall, there some words that suppose to be separated in the text but joined. For instance, Total "of436", "mitochondrial16S" etc. The authors should carefully read through the manuscript and correct them and others. 

Response: Thank you for this comment. We have addressed the problem accordingly.